# Rethinking On-policy Optimization for Query Augmentation

**Zhichao Xu**[1]                                                                                                       *zhichao.xu@utah.edu*

**Shengyao Zhuang**[2]

**Xueguang Ma**[3]

**Bingsen Chen**[4]

**Yijun Tian**[5]

**Fengran Mo**[6]

**Tao Li**[7]

**Jie Cao**[8]

**Vivek Srikumar**[1]                                                                                                      *svivek@cs.utah.edu*

[1]*University of Utah*      [2]*The University of Queensland*      [3]*University of Waterloo*      [4]*New York University*
[5]*University of Notre Dame*      [6]*Université de Montréal*      [7]*Google DeepMind*      [8]*University of Oklahoma*

**Reviewed on OpenReview:** *https://openreview.net/forum?id=mmqbjhz5Br*

## Abstract

Recent advances in large language models (LLMs) have led to a surge of interest in query augmentation for information retrieval (IR). Two main approaches have emerged. The first prompts LLMs to generate answers or pseudo-documents that serve as new queries, relying purely on the model's parametric knowledge or contextual information. The second applies reinforcement learning (RL) to fine-tune LLMs for query rewriting, directly optimizing retrieval metrics. While having respective advantages and limitations, the two approaches have not been compared under consistent experimental conditions. In this work, we present the first systematic comparison of prompting-based and RL-based query augmentation across diverse benchmarks, including evidence-seeking, ad hoc, and tool retrieval. Our key finding is that *under a compute-aware comparison setting, simple, training-free query augmentation often performs on par with, or even surpasses, more expensive RL-based counterparts, especially when using powerful LLMs.* Motivated by this discovery, we introduce a novel hybrid method, On-policy Pseudo-document Query Expansion (OPQE), in which the LLM policy learns to generate a pseudo-document that maximizes retrieval performance, rather than rewriting the query, thus merging the flexibility and generative structure of prompting with the targeted optimization of RL. We show OPQE outperforms both standalone prompting and RL-based rewriting, demonstrating that a synergistic approach yields the best results. We open source our implementation to facilitate reproducibility.[1]

## 1 Introduction

In many real-world retrieval scenarios, direct modification of the retriever is often not possible — for example, when relying on API-based services such as OpenAI's text-embedding-3 or proprietary systems like PubMed[2]. In such settings, improving retrieval effectiveness must come from the query side. One common approach is supervised fine-tuning (SFT), where a model is trained on human- or weakly-annotated pairs of original queries and rewritten (or clarified/expanded) queries (Yu et al., 2020; Mo et al., 2023). However, creating

---

[1]https://github.com/BaleChen/RethinkQAug
[2]https://pubmed.ncbi.nlm.nih.gov/

or obtaining high-quality query rewrite training data is nontrivial: humans may disagree about what makes a query "better", domain shifts can limit transfer, and annotated datasets are expensive to build (Yu et al., 2020; Ma et al., 2023; Coelho et al., 2025). Moreover, for dense retrievers the notion of what constitutes a "good" query is less well understood given their black-box neural architectures (Faggioli et al., 2023; Meng et al., 2023). Therefore, there has been growing research interest in unsupervised or weakly-supervised query augmentation methods, compared to data-centric SFT (Wu et al., 2022).

Large language models (LLMs) have been applied to query augmentation in two ways: (1) by rewriting queries via SFT using (query, rewritten query) pairs (or weak supervision) to improve query clarity or specificity; and (2) by rewriting or expanding queries in a zero- or few-shot manner, without additional training (Li et al., 2025). Prompt-based query augmentation methods such as HyDE (Gao et al., 2023) and Query2Doc (Wang et al., 2023) fall in the latter category: expanding queries with pseudo-documents generated from an LLM's parametric knowledge can significantly improve retrieval effectiveness. These zero- or few-shot approaches require no additional training and are simple to apply. However, prior works (Abe et al., 2025; Lei et al., 2024) have questioned these methods' robustness, as generating a fluent or seemingly relevant document does not always guarantee a high-quality query representation. LLMs' verbose outputs can introduce spurious terms, which may harm performance, particularly for sparse retrievers like BM25 (Robertson et al., 1995).

Beyond fine-tuning and prompting-based augmentation, another promising direction is to apply reinforcement learning (RL, Sutton et al., 1998), and in particular on-policy optimization, to query generation. Unlike methods that rely purely on prompt-based expansion, RL allows models to adapt their outputs based on retrieval-derived rewards, directly tying the generation process to ranking metrics such as recall, accuracy, or NDCG. This paradigm shifts the supervision signal from handcrafted query–rewrite pairs to feedback grounded in retrieval effectiveness.

Recent work such as DeepRetrieval (Jiang et al., 2025) represents one instantiation of this idea. It combines reasoning-style generation, inspired by frameworks like ReAct (Yao et al., 2023) and DeepSeek-R1 (Guo et al., 2025), with retrieval-based rewards to train LLMs that produce both intermediate reasoning steps and final queries. While DeepRetrieval demonstrates encouraging results across tasks including literature search, IR benchmarks, and text-to-SQL (Liu et al., 2025), the approach remains resource-intensive and dependent on relevance-labeled data. More broadly, the field still lacks a systematic comparison between prompting-based query augmentation methods and RL-based formulations, leaving open questions about their respective strengths, weaknesses, and applicability in real-world settings.

This question is increasingly central as RL-based search agents move into the mainstream. A recent line of work trains LLMs to interleave reasoning with retrieval through on-policy optimization against verifiable, retrieval-grounded rewards—Search-R1 (Jin et al., 2025), R1-Searcher (Song et al., 2025), and Re-Search (Chen et al., 2025a), among others (Lin et al., 2025b). In all of these systems, the policy's core action is to *formulate a query* to issue to a retriever; the surrounding agent loop simply decides when to search and how to use the results. Query augmentation is therefore the atomic retrieval action underlying agentic search, yet how best to optimize this action—and whether RL is even necessary to do so—remains poorly understood. We study this action in isolation, under controlled single-turn conditions, so that conclusions about it can inform the design of the larger agentic systems built on top of it (we survey this search-side line of work in Section 2).

In this work, we address this gap through a rigorous evaluation and benchmarking. Therefore, we ask: *under controlled and compute-aware comparisons, when does training-free pseudo-document generation suffice, and when does on-policy optimization provide meaningful benefits?*

To answer this question, we run a controlled comparison between prompting-based pseudo-document query expansion and on-policy RL for *direct query rewriting* (DeepRetrieval's formulation). Crucially, the two paradigms spend compute on different axes: RL invests *training-time* compute to specialize a small policy, whereas prompting spends *test-time* compute and can convert a fixed budget into a stronger inference model. This mirrors a broader question in the reasoning-LLM literature—whether test-time compute can substitute for, or even dominate, additional training (Snell et al., 2025; Kimi Team, 2025). We bring this question to query augmentation: accounting for these different compute profiles, we find that vanilla on-policy RL for *direct query rewriting* often yields limited gains, and is often outperformed by simple prompting backed by

a stronger model. Moreover, RL effects are strongly *task- and retriever-dependent*: improvements are more consistent for sparse term-matching retrievers such as BM25, while gains for dense retrievers can be smaller or even negative depending on the task and reward design. These observations motivate **O**n-policy **P**seudo-document **Q**uery **E**xpansion (**OPQE**), which applies on-policy optimization to *pseudo-document generation* rather than query rewriting, and yields more consistent improvements and the best overall performance.

Our contributions are threefold:

- We perform a controlled comparison of DeepRetrieval against a simple, zero-shot prompting baseline across diverse retrieval tasks, including classical ad hoc retrieval and the emerging agentic tool retrieval.

- Our experiments reveal a key finding: the simple, training-free prompting method often matches or surpasses more complex RL-trained methods. This observation suggests that leveraging the rich, structured knowledge of a powerful LLM via pseudo-document generation is a strong and resource-efficient baseline.

- We propose OPQE, a hybrid method that combines the generative structure of prompting with the targeted optimization of on-policy RL by training the policy to generate pseudo-documents under retrieval-based rewards, and it yields the strongest overall retrieval performance across our benchmarks.

## 2 Related Works

**Query Expansion and Reformulation.** Classical query expansion techniques include pseudo-relevance feedback (Rocchio, 1971; Voorhees, 1994), translation-based expansion (Cao et al., 2009), and statistical term weighting (Zhai & Lafferty, 2001). These approaches improve recall by introducing additional surface-level terms, but their effectiveness is often limited by noise sensitivity, especially for sparse retrievers.

More recently, LLMs have enabled zero- and few-shot query rewriting. Methods such as HyDE (Gao et al., 2023) and Query2Doc (Wang et al., 2023) prompt an LLM to generate pseudo-documents to be encoded by retrievers. Related approaches like ThinkQE (Lei et al., 2025) further incorporate reasoning before query generation. These methods are attractive because they are training-free and are directly applicable to API-based retrievers. However, their outputs are not explicitly optimized for retrieval metrics, and spurious terms may harm sparse retrieval performance (Abe et al., 2025; Lei et al., 2024). Refer to Li et al. (2025) for an in-depth survey.

**Learning to Rewrite Queries.** Beyond static prompting, several works explore training models for query reformulation. A prominent approach is supervised fine-tuning, where pre-trained sequence-to-sequence models like T5 (Raffel et al., 2020) are trained on datasets of original and rewritten queries (Wu et al., 2022; Liu et al., 2021). This technique is widely used in conversational search to transform ambiguous, context-dependent user utterances into clear, self-contained queries suitable for standard retrieval systems. However, the creation of high-quality, human-labeled datasets can be a significant bottleneck, and human rewrites may not always be optimal for retrieval performance (Wu et al., 2022; Ye et al., 2023). To address this weakness, recent efforts have turned to weakly supervised methods. One such strategy involves knowledge distillation, where a powerful large language model (LLM) acts as a "teacher" to generate a large synthetic dataset of rewritten queries. A smaller, more efficient "student" model is then fine-tuned on this data, learning the rewriting task without expensive human annotation (Ye et al., 2023). This methodology has proven effective within the "Rewrite-Retrieve-Read" framework for retrieval-augmented generation, where the initial query is adapted to be more effective for the downstream retrieval and reading components (Ma et al., 2023).

**RL for Query Optimization.** A growing line of work applies RL to optimize queries directly against retrieval-based rewards. Unlike supervised reformulation, which learns from a fixed dataset that requires extensive annotation, RL enables on-policy exploration, directly tying the training signal to retrieval effectiveness (Wang et al., 2020; Wu et al., 2022) and requires no golden (query, rewritten query) pairs. This is particularly valuable in dynamic contexts like conversational search, where an RL agent can be trained to reformulate queries to maximize the performance of a dense retriever (Zhu et al., 2025). This method's key advantage is its ability to optimize a rewriter for a black-box retrieval system, where the retriever's

parameters are inaccessible for fine-tuning. The rewriter is treated as a policy model, and it receives a reward based on the performance of the retriever, allowing it to adapt and generate more effective queries. While powerful, these methods can require substantial compute for exploration and often depend on carefully engineered reward functions to guide the learning process (Sheng et al., 2025; Lambert et al., 2024).

Within this family, a key design choice is whether to optimize *on-policy* or *off-policy*. On-policy methods such as REINFORCE (Sutton et al., 1998) and PPO (Schulman et al., 2017) update the policy using rollouts sampled from the current policy itself, scoring each sampled query against the live retriever. This couples exploration directly to retrieval reward and requires no pre-collected supervision, but it is comparatively expensive: every gradient step depends on fresh generations and retrieval calls. Off-policy alternatives instead learn from a fixed dataset of trajectories or preference pairs collected ahead of time; in the query-generation setting this is most naturally realized as preference tuning, e.g., aligning a query generator with ranking-derived preferences via Direct Preference Optimization and its variants (Rafailov et al., 2023; Park et al., 2024; Meng et al., 2024; Coelho et al., 2025). Such methods avoid repeated online retrieval but inherit the cost and coverage limitations of constructing the offline preference data, and they optimize a surrogate preference objective rather than the retrieval metric directly. Because our goal is to study query augmentation that ties the training signal as closely as possible to retrieval effectiveness without assuming any pre-collected rewrite or preference data, we focus on on-policy optimization, and in particular PPO, throughout this work (see Section 3).

**Search-Side Optimization for Agentic QA.** A rapidly growing body of work improves the *search side* of RAG and deep-research agents, and it broadly splits into two strategies. The first strengthens the retrieval stack itself: ReasonIR (Shao et al., 2025c) trains a retriever specifically for reasoning-intensive queries, while Meng et al. (2026) revisit text ranking in the deep-research setting and show that re-ranking substantially improves agent performance, also observing that agent-generated queries carry a particular surface form that favors certain retrievers. The second strategy improves how the agent *orchestrates* queries: ParallelSearch (Zhao et al., 2025b) and ExpandSearch (Zhao et al., 2025a) train policies to decompose a query into multiple sub-queries searched in parallel and then aggregate the results, HybridDeepSearcher (Ko et al., 2025) interleaves such parallel expansion with sequential reasoning, and Chroma's Context-1 (Bashir et al., 2026) acts as a retrieval subagent that decomposes a high-level goal into iterative multi-turn searches while self-editing its context to discard irrelevant results. These directions are complementary to ours: they optimize the retriever/reranker or the multi-query control loop, whereas we study how best to formulate the *individual* query-augmentation action — the atomic unit that every one of these systems issues, aggregates, or re-ranks. Our compute-aware comparison of prompting vs. on-policy RL for this action, and the OPQE formulation, are therefore orthogonal to and composable with these search-side advances.

Closest to our setting is the concurrent Rec-R1 (Lin et al., 2025a), which applies RL with retrieval-metric rewards to recommendation and finds that the *format* of generated text must align with the retriever's training distribution — dense retrievers trained on user review–product pairs improve substantially when queries are rewritten into review-style text rather than standard rewrites. This resonates with our pseudo-document findings: generating document-style text better matches the input distribution of dense retrievers trained on passage-level contrastive objectives. Whereas Rec-R1 studies a domain where RL consistently beats prompting, we show that for general IR tasks prompting with capable LLMs is competitive under compute-aware settings. Overall, our work conducts a rigorous empirical comparison of prompting- vs. RL-based query augmentation, highlighting their trade-offs across sparse and dense retrieval where retrievers are treated as black boxes, and contributes OPQE as a hybrid that combines the pseudo-document format with RL optimization.

## 3 Background

**Prompt-based Query Augmentation.** Query augmentation aims to rewrite or expand an original query to improve retrieval effectiveness. Sparse methods such as BM25 (Robertson et al., 1995) benefit directly from term-level expansions, while dense retrievers, which embed queries and documents into a shared vector space, pose a greater challenge since it is unclear what constitutes an effective query embedding. Collecting human-labeled training data for this task is expensive and often infeasible.

To reduce the dependency on labeled data, recent methods exploit the zero- and few-shot generative capabilities of LLMs. A common approach is to prompt an LLM to generate a pseudo-document that might answer the query. This pseudo-document is then encoded by the retriever to perform retrieval. Such prompt-based pseudo-document expansion avoids the need for explicit query-rewrite supervision and has been shown to be effective across open-domain QA and web search benchmarks (Gao et al., 2023; Wang et al., 2023). However, its success is tightly coupled to the scale and internal knowledge of the LLM, raising questions about how to enable smaller models to serve as effective query rewriters.

**RL-based Query Augmentation.** Another approach is to optimize query rewriting via RL. Here, the query rewriter is treated as a policy $\pi_\theta(q' \mid q)$ that generates an augmented query $q'$ given the original query $q$. The process can be formalized as a Markov Decision Process (MDP), where **State** $s_t$ denotes the current context, typically the original query $q$ and the intermediate reasoning steps. **Action** $a_t$ means generating the next token or sub-query in $q'$. **Policy** $\pi_\theta(a_t \mid s_t)$ denotes LLM parameterized by $\theta$. **Reward** $r(s_t, a_t)$ is derived from retrieval effectiveness of the completed query $q'$, e.g., Recall@K, NDCG. The objective is to maximize the expected retrieval reward, with a KL-regularization term relative to a reference policy $\pi_{\text{ref}}$ to stabilize training and maintain generation fluency:

$$\mathcal{J}_{\text{KL}}(\theta) = \mathbb{E}_{q' \sim \pi_\theta(\cdot \mid q)} \Big[ r_{\text{retrieval}}(q') - \beta \, \text{KL}\big(\pi_\theta(\cdot \mid q) \, \| \, \pi_{\text{ref}}(\cdot \mid q)\big)\Big]. \tag{1}$$

This objective can be optimized with on-policy RL methods such as Proximal Policy Optimization (PPO, Schulman et al., 2017).

A useful way to situate this formulation is through the lens of reinforcement learning with verifiable rewards (RLVR), the paradigm behind recent reasoning models such as DeepSeek-R1 (Guo et al., 2025) and Tülu 3 (Lambert et al., 2024). RLVR replaces a learned reward model with an automatically checkable signal — e.g., whether a math answer is correct or code passes its tests — which both stabilizes training and avoids reward hacking. Retrieval-based query augmentation is naturally an instance of RLVR: the reward $r_{\text{retrieval}}(q')$ is computed by running a fixed retriever against ground-truth relevance labels (e.g., Recall@K, NDCG, or Completeness@K), so the verifier is the retriever–corpus environment together with its relevance judgments rather than a trained critic. This places query augmentation in the same methodological family as reasoning-focused RLVR, with the retriever supplying the verification signal.

**Reasoning-Style Query Generation.** Recent works have demonstrated the benefit of generating structured outputs, separating intermediate reasoning from the final answer. Frameworks like ReAct (Yao et al., 2023) and DeepSeek-R1 (Guo et al., 2025) inspire approaches where the policy first generates reasoning steps $h_1, \ldots, h_n$ and then a final augmented query $q'_{\text{final}}$. The retrieval reward is computed against $q'_{\text{final}}$, while intermediate steps may be guided via auxiliary heuristics or shaping rewards. DeepRetrieval (Jiang et al., 2025) exemplifies this paradigm by combining reasoning-style generation with PPO training on retrieval-based rewards. While effective, this approach highlights general challenges in RL-based query augmentation: reward sparsity, exploration vs. fluency trade-offs, and computational cost for large LLMs.

**Terminology and Scope.** We deliberately scope this work to *on-policy* optimization rather than reinforcement learning in its broadest sense, and the distinction is reflected in our title. As discussed in Section 2, on-policy methods (e.g., PPO (Schulman et al., 2017)) optimize directly against the live retrieval reward in Eq. (1), whereas the off-policy alternatives most relevant here amount to *preference tuning* over a fixed offline dataset (Rafailov et al., 2023; Coelho et al., 2025) and optimize a surrogate objective. Because these paradigms differ substantially in their assumptions and cost profiles, a broad claim about "RL for query augmentation" would overclaim relative to the experiments we actually run. We therefore restrict our study, and our conclusions, to on-policy optimization — specifically PPO — and use the term *RL* in the remainder of the paper as shorthand for this on-policy setting unless otherwise noted. We leave a systematic comparison against off-policy preference-tuning methods to future work.

**Algorithm Overview.** The procedure for RL-based query augmentation can be summarized as Algorithm 1. This formulation allows training a rewriter even when the retrieval system is a black-box, such as an API-based service, emphasizing the connection between query generation and retrieval performance.

---

**Algorithm 1** Direct query rewriting with PPO

---

1: Initialize LLM policy $\pi_\theta$, reference policy $\pi_{\mathrm{ref}}$, and value function $V_\phi$
2: **for** each batch $\mathcal{B}$ from dataset **do**
3:     Initialize rollout buffer $\mathcal{D} \leftarrow \emptyset$
4:     **for** each query $q \in \mathcal{B}$ **do**
5:         Sample a rollout (reasoning + augmented query): $\tau \sim \pi_\theta(\cdot \mid q)$
6:         Extract augmented query $q'$ from $\tau$
7:         Retrieve top-$k$ passages with retriever
8:         Compute retrieval reward $r_{\mathrm{retrieval}}(q')$ with potential format constraints
9:         Assign the scalar reward to the rollout $\tau$
10:        Estimate sequence-level advantages $\hat{A}(\tau)$ using $V_\phi$ (e.g., GAE)
11:        Add $(\tau, r_{\mathrm{retrieval}}(q'), \hat{A}(\tau))$ to $\mathcal{D}$
12:     **end for**
13:     **for** each PPO epoch $e = 1, \ldots, E$ **do**
14:         **for** each minibatch $\mathcal{M} \subset \mathcal{D}$ **do**
15:             Update $\theta$ using PPO clipped objective + KL penalty to $\pi_{\mathrm{ref}}$
16:             Update $\phi$ by regression on returns (value loss)
17:         **end for**
18:     **end for**
19: **end for**

---

## 4 Comparison

We perform a systematic empirical comparison of on-policy optimization for query augmentation — exemplified by DeepRetrieval (Jiang et al., 2025) — against a simple prompt-based method (**SPQE**, **S**imple **P**seudo-document **Q**uery **E**xpansion). Our goal is to quantify the relative benefits, limitations, and trade-offs of RL-based and zero-shot query augmentation approaches across different retrieval scenarios.

### 4.1 Methodology

We begin with a rigorous replication of DeepRetrieval's experiments. Our setup mirrors the original work, including both sparse and dense retrieval models, where the retriever and corpus together form the "environment" in RL terminology. For sparse retrieval, we use BM25 (Robertson et al., 1995), and for dense retrieval, we use task-appropriate retrievers, as detailed in Section 4.2. In tables, we denote the DeepRetrieval baselines as DR-3B and DR-7B based on the policy size.

**Replication Checklist.** To ensure faithful replication, we follow DeepRetrieval's overall formulation: (i) the policy generates intermediate reasoning followed by a final augmented query in a fixed format (Appendix Section E); (ii) the retrieval reward is computed from the final query against a fixed retriever–corpus environment; (iii) the overall reward includes a format constraint and a retrieval component (Appendix Section C); and (iv) we optimize with PPO under a KL-regularized objective (Algorithm 1). We only modify engineering details to improve throughput (e.g., environment implementation), while keeping the task definition, reward, and evaluation metrics aligned with the original setup.

For the prompt-based method (SPQE), we instruct an instruction-following LLM to generate a hypothetical document $d^H$ addressing the query $q$, and use the concatenated $(q, d^H)$ as the augmented query. This zero-shot approach differs from Query2Doc (Wang et al., 2023), which relies on few-shot prompting. Compared to the similar zero-shot method HyDE (Gao et al., 2023), SPQE skips the follow-up steps of embedding multiple pseudo-documents and aggregating their representations, and is designed to function as a simple, strong baseline for zero-shot query augmentation.

For RL-based query augmentation, we follow DeepRetrieval's reward design: the policy first generates reasoning steps, then output the enhanced query. Detailed reward formulations are provided in Appendix Section C.

## 4.2 Experimental Setup

**Datasets, Retrieval and Evaluations.**   We follow the experimental setup of Jiang et al. (2025) to include evidence-seeking retrieval and ad hoc retrieval datasets.  For evidence-seeking retrieval, we use Wikipedia-18[3] as the retrieval corpus, and include three query collections: NQ (Kwiatkowski et al., 2019), TriviaQA (Joshi et al., 2017) and SQuAD (Rajpurkar et al., 2016).  For ad hoc retrieval, we use following datasets from BEIR (Thakur et al., 2021): FEVER (Thorne et al., 2018), HotpotQA (Yang et al., 2018), NFCorpus (Boteva et al., 2016) and SciFact (Wadden et al., 2020), as these datasets include training sets for on-policy optimization.  We also include evaluation on MS MARCO's dev set with shallow annotation depth (Bajaj et al., 2016), alongside DL19 (Voorhees et al., 2020) and DL20 (Craswell et al., 2025) with in-depth annotations. Refer dataset statistics to Appendix Section A.

For evidence-seeking retrieval, we follow Ma et al. (2021) to measure Hit@20 (answer span hits).  We use BM25 as the sparse retriever and E5-base-v2 (Wang et al., 2022) as the dense retriever.

For ad hoc retrieval, we use NDCG@10 as the evaluation metric following Thakur et al. (2021).  For retrieval, we use their respective corpus, and use BM25 for sparse retrieval and Contriever-msmarco (Izacard et al., 2022) as the dense retriever.  Note that Jiang et al. (2025) use different dense retrievers for different ad hoc retrieval datasets, but we control the dense retriever model.

**Backbone Models.**   We use Qwen2.5-{3B,7B}-Instruct (Qwen, 2024) as our initial policy.  For implementation of SPQE, we use different LLMs. Our main experimental result is based on proprietary GPT-4o-mini (Menick et al., 2024) and we also report results with open-weight GPT-OSS-120B (Agarwal et al., 2025) and Qwen3-32B-Instruct (Yang et al., 2025). *Importantly, we intentionally allocate more inference-time compute to prompting by using larger LLMs, so that the overall compute budget is closer to that of on-policy RL training (which uses smaller base policies but incurs substantial training-time compute).* As a result, our SPQE baselines are not backbone-matched to the DR-3B/7B policies; in preliminary experiments, SPQE with the same Qwen2.5-{3B,7B}-Instruct backbones was not as competitive, as they are not directly trained for the query rewriting tasks.  Specifically, our PPO training runs use 8x A100 40GB GPUs for 3B policies and 16x A100 40GB GPUs for 7B policies, whereas SPQE only incurs inference cost.  Note that SPQE incurs per-query inference cost, while RL incurs an up-front multi-GPU training cost that is amortized over the evaluation set.  Therefore, their wallclock costs are not directly comparable.  In practice, when per-dataset RL training is infeasible (e.g., due to compute or frequent domain shifts), a zero-shot prompting baseline like SPQE can serve as a strong default.

To reduce the possibility that SPQE performance is driven solely by access to proprietary LLMs, we additionally evaluate SPQE with strong open-weight instruction models (Qwen3-32B-Instruct and GPT-OSS-120B). As shown in Table 2, these open-weight models achieve performance that is generally comparable to GPT-4o-mini across evidence-seeking, ad hoc, and tool retrieval settings (full per-dataset results in Appendix Section B). Notably, SPQE with an open-weight 32B model can be served on a single A100 40GB GPU (e.g., using vLLM).

**Implementation Details.**   We use the original codebase from Jiang et al. (2025) and made several modifications to improve training throughput.  Our retrieval environment is based on Pyserini (Lin et al., 2021) and Faiss (Johnson et al., 2019). We use veRL 0.1 (Sheng et al., 2025) as the RL framework and PPO (Schulman et al., 2017) as the specific on-policy RL algorithm, matching DeepRetrieval's PPO-based training setup. We did not include group-based on-policy RL algorithms such as GRPO (Shao et al., 2024; Guo et al., 2025); while such alternatives may yield similar performance in single-turn retrieval environments with verifiable rewards, we do not evaluate them here and leave a systematic comparison of on-policy algorithms to future work. Without otherwise specified, we use learning rate 1e-6, global train batch size 128, actor and critic mini batch size 32, rollout temperature 0.6.  All experiments are performed on 8x or 16x Nvidia A100 GPUs, each with 40GB VRAM.

---

[3]The 2018-12-20 snapshot of English Wikipedia, chunked into non-overlapping 100-word passages (Ma et al., 2021)

Table 1: Comparison of model performance on evidence-seeking retrieval and ad hoc retrieval datasets. Sparse columns use BM25 retriever; dense columns use E5-base-v2 for evidence-based retrieval datasets and Contriever-msmarco for ad hoc retrieval datasets. Best overall numbers are **bold**, best in-category numbers are underlined.

| | *Sparse BM25 Retriever* | | | | | | *Dense Retriever* | | | | | |
|---|---|---|---|---|---|---|---|---|---|---|---|---|
| Dataset | Base | DR-3B | DR-7B | SPQE-3B | SPQE-7B | SPQE | Base | DR-3B | DR-7B | SPQE-3B | SPQE-7B | SPQE |
| *Evidence-seeking Datasets (Dense = E5-base-v2)* | | | | | | | | | | | | |
| NQ | 63.0 | 73.2 | 73.8 | 70.8 | 73.8 | 80.7 | **86.1** | 85.8 | 85.6 | 81.8 | 82.1 | 85.2 |
| TriviaQA | 76.4 | 79.8 | 80.9 | 78.0 | 80.1 | 84.1 | 83.5 | 82.8 | 83.7 | 81.2 | 81.7 | **85.0** |
| SQuAD | 71.1 | 72.1 | **72.2** | 66.3 | 67.9 | 69.7 | 70.2 | 70.3 | 70.2 | 65.8 | 66.3 | 68.8 |
| Average Evidence-seeking | 70.2 | 75.0 | 75.6 | 71.7 | 73.9 | 78.2 | **79.9** | 79.6 | 79.8 | 76.3 | 76.7 | 79.7 |
| *Ad Hoc Retrieval Datasets (Dense = Contriever-msmarco)* | | | | | | | | | | | | |
| MS MARCO | 22.8 | 23.5 | 24.1 | 19.7 | 19.1 | 21.9 | 40.7 | 40.7 | **40.8** | 30.3 | 30.0 | 31.9 |
| DL19 | 50.6 | 50.8 | 51.3 | 53.8 | 57.3 | 60.0 | 67.5 | 67.7 | 67.8 | 61.3 | 64.1 | **70.1** |
| DL20 | 48.0 | 49.4 | 49.9 | 51.7 | 53.4 | 60.1 | 66.6 | 66.6 | 66.5 | 62.3 | 65.1 | **69.3** |
| NFCorpus | 32.5 | 34.3 | **35.9** | 31.6 | 31.6 | 33.9 | 32.8 | 33.5 | 34.2 | 32.7 | 34.3 | 34.3 |
| HotpotQA | 63.3 | 64.5 | 65.2 | 56.4 | 57.4 | 66.5 | 63.8 | 65.0 | 64.9 | 59.5 | 60.1 | **68.2** |
| FEVER | 65.1 | 72.3 | 74.7 | 73.2 | 74.2 | 79.8 | 75.8 | **85.5** | 84.8 | 81.2 | 82.3 | 83.9 |
| SciFact | 66.5 | 66.9 | 66.2 | 69.3 | 69.7 | **71.4** | 67.7 | 66.3 | 66.9 | 66.3 | 67.5 | 68.9 |
| FiQA | 23.6 | 25.6 | 24.7 | 17.3 | 19.0 | 19.5 | 32.9 | 33.1 | **34.0** | 24.0 | 28.1 | 26.5 |
| Average Ad Hoc Retrieval | 46.6 | 48.4 | 49.0 | 46.6 | 47.7 | 51.6 | 56.0 | 57.3 | **57.5** | 52.2 | 53.9 | 56.6 |

Table 2: Prompting (SPQE) with proprietary vs. open-weight LLMs, compared against RL-trained policies (averaged over datasets). Evidence-seeking uses Hit@20, ad hoc retrieval uses NDCG@10, and tool retrieval uses Recall@10. Full per-dataset results for SPQE are provided in Appendix Section B.

| Task family | Retriever | Base | GPT-4o-mini | Qwen3-32B-Instruct | GPT-OSS-120B | DR-3B | DR-7B |
|---|---|---|---|---|---|---|---|
| Evidence-seeking | BM25 | 70.2 | 78.2 | 76.5 | **80.0** | 75.0 | 75.6 |
| Evidence-seeking | E5-base-v2 | **79.9** | 79.7 | 78.5 | 79.7 | 79.6 | 79.8 |
| Ad hoc (BEIR) | BM25 | 46.6 | **51.6** | 50.8 | 51.3 | 48.4 | 49.0 |
| Ad hoc (BEIR) | Contriever | 56.0 | 56.6 | 56.2 | 51.8 | 57.3 | **57.5** |
| Tool retrieval | BM25 | 39.9 | **44.4** | 44.1 | 44.3 | 41.4 | 39.8 |
| Tool retrieval | Contriever | 38.8 | 44.8 | **45.4** | 44.3 | 38.6 | 37.8 |
| *Overall* | | | | | | | |
| Borda score | Sparse | 4.0 | **17.0** | 12.0 | 16.0 | 7.0 | 7.0 |
| Borda score | Dense | 11.0 | **12.5** | 10.0 | 8.5 | 9.0 | 12.0 |
| Borda score | All | 15.0 | **29.5** | 22.0 | 24.5 | 16.0 | 19.0 |
| Borda rank | All | 6 | **1** | 3 | 2 | 5 | 4 |

## 4.3 Result and Analysis

We show the results of evidence-seeking retrieval and ad hoc retrieval in Table 1. Overall, both on-policy optimization and simple prompt-based expansion (SPQE) improve retrieval performance over not using any query augmentation (the "Base" columns), but their benefits differ across tasks and retrievers.

On-policy optimization can be effective, especially when training smaller policies. For ad hoc retrieval with a dense retriever, the DR-7B (DeepRetrieval) model achieves the highest average score of 57.5, improving over the base retriever's score of 51.6; this is particularly noticeable on datasets like FEVER, where the score increases from 75.8 to 85.5. Similarly, with a sparse BM25 retriever on evidence-based datasets, the DR-7B model improves the average score from 70.2 to 75.6.

Furthermore, Table 1 shows that our simpler prompt-based method, SPQE, is also highly effective and competitive. For sparse retrieval, SPQE consistently delivers the best performance, achieving the highest average scores for both evidence-based (78.2) and ad hoc (51.6) datasets.

To provide a consolidated view across sparse vs. dense settings (and to disentangle backbone effects for prompting), we summarize prompting with proprietary and open-weight LLMs alongside the DeepRetrieval

Table 3: Comparison of model performance on tool retrieval datasets. Sparse use BM25 retriever; dense use Contriever-msmarco retriever. Best overall numbers on each dataset are **bold**, best in-category numbers are underlined.

| Dataset | NDCG@10 | | | | Recall@10 | | | | Completeness@10 | | | |
|---|---|---|---|---|---|---|---|---|---|---|---|---|
| | Base | DR-3B | DR-7B | SPQE | Base | DR-3B | DR-7B | SPQE | Base | DR-3B | DR-7B | SPQE |
| *Sparse BM25 Retriever* | | | | | | | | | | | | |
| ToolRet-Web | 24.8 | 28.8 | 26.7 | 27.4 | 33.6 | 38.3 | 35.9 | 38.0 | 26.0 | 29.7 | 27.9 | 30.2 |
| ToolRet-Code | 30.1 | 30.1 | 29.8 | **33.4** | 44.3 | 41.6 | 41.5 | **46.0** | 37.1 | 38.9 | 38.3 | **42.7** |
| ToolRet-Customized | 35.0 | 37.3 | 35.4 | **39.6** | 41.7 | 44.3 | 41.9 | **49.4** | 24.9 | 28.2 | 26.4 | **32.4** |
| Average | 30.0 | 32.0 | 30.7 | 33.4 | 39.9 | 41.4 | 39.8 | 44.4 | 29.4 | 32.3 | 30.8 | **35.1** |
| *Dense Contriever-msmarco Retriever* | | | | | | | | | | | | |
| ToolRet-Web | 31.3 | 31.6 | 31.1 | **34.2** | 41.1 | 41.5 | 40.9 | **45.1** | 31.6 | 32.0 | 31.4 | **35.0** |
| ToolRet-Code | 24.5 | 24.4 | 24.5 | 32.2 | 33.4 | 33.2 | 33.5 | 42.2 | 30.5 | 30.5 | 30.8 | 38.3 |
| ToolRet-Customized | 33.9 | 33.4 | 31.3 | 37.1 | 41.8 | 41.0 | 38.8 | 47.2 | 26.8 | 27.1 | 26.1 | 31.2 |
| Average | 29.9 | 29.8 | 29.0 | **34.5** | 38.8 | 38.6 | 37.8 | **44.8** | 29.6 | 29.9 | 29.4 | 34.8 |

baselines in Table 2. Across all six task–retriever combinations, prompting achieves the best overall Borda rank (rank 1), while the DeepRetrieval policies rank behind the prompting variants (rank 4 for DR-7B and rank 5 for DR-3B). This aggregate result reinforces the main takeaway from Table 1: under our compute-aware setup (larger inference-time models for prompting vs. smaller RL-trained policies with substantial training cost), simple pseudo-document prompting is a very strong baseline and achieves the best overall average ranking in our experiments.

For dense retrieval, prompting performance is also close to (and sometimes surpasses) the more complex RL method, while RL can still be beneficial on specific datasets (e.g., FEVER and FiQA) when adapted to a particular retriever and domain. We defer a more in-depth discussion about SPQE vs. on-policy optimization to Section 4.5.

## 4.4 Extending to Tool Retrieval

Retrieving useful tools from tool sets for LLM agents is becoming increasingly critical in real-world applications. However, the main challenge is that candidate tools are usually large-scale and many of them are similar in functionality (Patil et al., 2024; Qin et al., 2024; Qu et al., 2024; Shi et al., 2025). How query augmentation might enhance tool retrieval efficacy is rather underexplored due to the lack of large-scale training and evaluation datasets. We address the gap by studying prompting-based and RL-based query augmentation in the context of tool retrieval.

**Datasets.** We use a recently released tool retrieval dataset ToolRet (Shi et al., 2025), which is constructed from existing tool learning datasets. The training set contains over 200k instances, which allows us to study on-policy RL training at scale.

**Training and Evaluation.** We construct the training set and training corpus from the released training set. Note that here the training corpus (i.e., environment) is different from the corpus we run evaluation on, due to the mismatch of candidate tools. We evaluate on three official splits: ToolRet-Web, ToolRet-Code, ToolRet-Customized. We train the same base policy as Section 4.2: Qwen2.5-{3B, 7B}-Instruct and adopt BM25 and E5-base-v2 as the retrievers. We report performance of SPQE as a simple prompting-based method compared to on-policy optimization. For evaluation, we focus on classical retrieval metrics NDCG@10 and Recall@10. We further include Completeness@10. The Completeness metric was proposed by Qu et al. (2024) to measure whether all the required tools have been included in the ranklist. For reward design, we directly optimize against the Completeness@10 metric. We leave further investigation of fine-grained reward design to future works.

**Results and Analysis.** We show the results of tool retrieval in Table 3. Similar to our prior results, we observe that query augmentation methods can substantially improve performance over the baseline. On-

policy optimization shows some benefits, particularly for the sparse retriever, but also reveals a notable disparity in its effectiveness between sparse and dense models.

For Completeness@10, the metric we optimized for, the DR-3B (DeepRetrieval) model improves the average score for the sparse BM25 retriever from a base of 29.4 to 32.3. This demonstrates that RL can successfully steer the model to improve a target metric. However, for the dense retriever, the same RL training shows no benefit, with average Completeness@10 scores (29.9 for DR-3B, 29.4 for DR-7B) failing to surpass the base score of 29.6. This disparity between sparse and dense retriever is critical. While DeepRetrieval provides a modest lift for BM25 on both NDCG@10 (from 30.0 to 32.0) and Completeness@10, it consistently degrades performance for the dense Contriever across all metrics. For instance, the DR-7B model lowers the average NDCG@10 from 29.9 to 29.0. This observation suggests that the queries generated by the RL policy, while optimized for the Completeness reward, may be producing keyword-focused outputs that help term-matching systems like BM25 but result in less effective embeddings for semantic-focused dense retrievers. A more direct diagnosis would require analyzing model outputs (e.g., length/verbosity, term distribution shifts, and query–tool coverage); we do not include these statistics in this paper and leave them to future work.

In contrast, the simple prompting method SPQE is consistently strong across both sparse and dense tool retrieval settings. It achieves the highest average scores for both NDCG@10 (33.4 for sparse, 34.5 for dense) and Completeness@10 (35.1 for sparse, 34.8 for dense). Furthermore, looking at Recall@10, HyDE, another generative approach, consistently outperforms all other methods. This indicates that for the task of tool retrieval, generating richer, document-like queries via simple prompting can be a reliable strategy for improving performance on both sparse and dense systems relative to the specific RL formulation tested here.

### 4.5 Discussions

Our experiments show that both on-policy optimization and prompt-based expansion improve retrieval, but they trade off between targeted optimization and general applicability.

On-policy optimization excels at adapting a model to a specific retriever and domain. This is most evident on the specialized FiQA dataset (Table 1), where the RL-trained model outperforms prompting, likely by learning to generate queries with domain-specific financial terminology. However, this approach is computationally expensive, requires labeled data for rewards, and may not always achieve performance improvement. For instance, our ToolRetrieval policy underperforms no query augmentation setting with dense retrievers.

In contrast, prompt-based expansion (e.g., SPQE) is a robust and resource-efficient baseline that performs well across nearly all settings. We hypothesize that its effectiveness stems from three factors. First, it leverages the LLM's internal knowledge to expand queries with relevant terms, mitigating the vocabulary mismatch problem for sparse retrievers like BM25; this is consistent with the strong zero-shot BM25 gains in Table 1 and the higher initial rewards when training OPQE (see Fig. 1). Second, pseudo-documents provide *richer semantic specification* of an otherwise underspecified query: a short query like "when did X happen" conveys limited information, whereas a generated passage adds contextual terms that help a learned encoder produce a more discriminative embedding (Xu et al., 2026a). This is further aided by *better alignment with the retriever's input distribution*: learned retrievers are often trained on passage-level text via contrastive learning, so a pseudo-document is structurally closer to the retriever's training distribution than a short query. Third, the LLM's parametric knowledge fills in terms and context that the original query lacks, providing both sparse and dense retrievers with more signal to work with (Xu et al., 2026c).

**Cost Trade-offs.** The two paradigms also differ in their deployment cost profiles. RL incurs a substantial up-front training cost that is amortized over all future queries: once trained, the policy can be served with a small model (e.g., 7B parameters). Prompting with a stronger LLM incurs a per-query inference cost that scales linearly with query volume. In high-throughput, fixed-corpus scenarios where a dedicated policy can be trained once and deployed indefinitely, RL's amortized cost may be lower. Conversely, in scenarios with frequent domain shifts, limited compute infrastructure, or diverse corpora where per-corpus RL training is impractical, zero-shot prompting is more practical. Our paper does not claim one paradigm is universally preferable; rather, we characterize when each is advantageous.

Table 4: Comparison of model performance on evidence-based retrieval and ad hoc retrieval datasets. We apply OPQE on 3B and 7B models. Sparse columns use BM25 retriever; dense columns use E5-base-v2 for evidence-based retrieval datasets and Contriever-msmarco for ad hoc retrieval datasets. Best overall numbers are **bold**, best in-category numbers are underlined.

| Dataset | *Sparse BM25 Retriever* | | | | | | *Dense Retriever* | | | | | |
|---|---|---|---|---|---|---|---|---|---|---|---|---|
| | Base | DR-3B | DR-7B | SPQE | OPQE-3B | OPQE-7B | Base | DR-3B | DR-7B | SPQE | OPQE-3B | OPQE-7B |
| *Evidence-based Retrieval Datasets (Dense = E5-base-v2)* | | | | | | | | | | | | |
| NQ | 63.0 | 73.2 | 73.8 | 80.7 | 74.7 | 75.2 | **86.1** | 85.8 | 85.6 | 85.2 | 85.9 | 85.9 |
| TriviaQA | 76.4 | 79.8 | 80.9 | 84.1 | 80.6 | 81.0 | 83.5 | 82.8 | 83.7 | **85.0** | 83.3 | 83.3 |
| SQuAD | 71.1 | 72.1 | 72.2 | 69.7 | **73.3** | 73.2 | 70.2 | 70.3 | 70.2 | 68.8 | 71.4 | 70.1 |
| Average | 70.2 | 75.0 | 75.6 | 78.2 | 76.2 | 76.5 | 79.9 | 79.6 | 79.8 | 79.7 | **80.2** | 79.8 |
| *Ad Hoc Retrieval Datasets (Dense = Contriever-msmarco)* | | | | | | | | | | | | |
| MS MARCO | 22.8 | 23.5 | 24.1 | 21.9 | 23.8 | 25.1 | 40.7 | 40.7 | **40.8** | 31.9 | 40.5 | 37.5 |
| DL19 | 50.6 | 50.8 | 51.3 | 60.0 | 53.9 | 56.6 | 67.5 | 67.7 | 67.8 | 70.1 | 67.8 | 69.5 |
| DL20 | 48.0 | 49.4 | 49.9 | 60.1 | 51.3 | 55.4 | 66.6 | 66.6 | 66.5 | 69.3 | 66.8 | **70.5** |
| NFCorpus | 32.5 | 34.3 | **35.9** | 33.9 | **35.9** | 35.7 | 32.8 | 33.5 | 34.2 | 34.3 | 33.5 | 34.6 |
| HotpotQA | 63.3 | 64.5 | 65.2 | 66.5 | 63.3 | 65.4 | 63.8 | 65.0 | 64.9 | **68.2** | 63.7 | 64.9 |
| FEVER | 65.1 | 72.3 | 74.7 | 79.8 | 81.1 | 80.7 | 75.8 | **85.5** | 84.8 | 83.9 | 84.9 | 85.4 |
| SciFact | 66.5 | 66.9 | 66.2 | **71.4** | 67.9 | 68.6 | 67.7 | 66.3 | 66.9 | 68.9 | 67.9 | 68.4 |
| FIQA | 23.6 | 25.6 | 24.7 | 19.5 | 25.0 | 24.9 | 32.9 | 33.1 | 34.0 | 26.5 | 34.1 | **34.4** |
| Average | 46.6 | 48.4 | 49.0 | 51.6 | 50.3 | 51.6 | 56.0 | 57.3 | 57.5 | 56.6 | 57.4 | **58.1** |

**Elicitation vs. Acquisition.** Our results also speak to an ongoing debate about what on-policy RL actually contributes. Recent analyses of RLVR for reasoning argue that RL primarily *elicits* and sharpens capabilities already latent in the base model, rather than *acquiring* genuinely new ones (Yue et al., 2026; Wang et al., 2026; Shao et al., 2025b). Our findings are consistent with this view in the retrieval setting. A capable base model prompted to generate a pseudo-document is already a strong query augmenter, and on-policy optimization over direct query rewriting often fails to exceed it; the gains RL does provide are concentrated where the base policy is weak (smaller models) or where the retriever rewards a specific surface form (e.g., term-matching for BM25, or domain-specific vocabulary on FiQA). The training trajectories of OPQE make this concrete: framing the action as pseudo-document generation gives the policy a high-reward "warm start" (Fig. 1), and RL then refines an already-effective behavior rather than learning it from scratch. From this perspective, the contribution of RL here is less about teaching the model what a good query looks like and more about adapting a capability it largely already has to the idiosyncrasies of a particular retriever.

# 5 On-policy Pseudo-document Query Expansion

Our experiments in Section 4.5 highlighted a key difference between two paradigms: on-policy RL methods rewrite the original query $q \rightarrow q'$, while prompt-based expansion methods like SPQE generate a pseudo-document $d^H$ to provide richer, more structured information to the retriever. This observation motivates a natural follow-up question: *can we combine the strengths of both approaches to further improve query augmentation?* In this section, we introduce a straightforward modification to achieve this synergy.

## 5.1 Method

We make a simple modification to the DeepRetrieval style on-policy optimization RL process for direct query rewriting. Instead of instructing the policy model to rewrite the query $q \rightarrow q'$, we instruct the model to write a hypothetical document $d^H$, then directly output the concatenated $(q, d^H)$ (prompt template in Appendix Section E). This reformulation inherently changes the output structure and typical length distribution of the augmented query compared to direct query rewriting. Then this concatenated string is used to retrieve relevant documents from the environment, similar to SPQE, and the reward is computed by the retrieval metrics like NDCG. In this way, we merge the SPQE into the on-policy optimization process. We refer to this method as **O**n-policy **P**seudo-document **Q**uery **E**xpansion (**OPQE**).

**Prompt Design.** The key difference from DeepRetrieval's prompt template lies in the task instruction: DR instructs the policy to "create query terms" or "expand [the query] with additional semantically relevant information" (Appendix Figs. 3 and 4), whereas OPQE instructs the policy to "write a concise Wikipedia-style passage that answers the query" (Appendix Fig. 6). Both templates share the same reasoning-then-answer structure with `<think>` and `<answer>` tags. Crucially, the OPQE output format includes the original query followed by the generated passage (separated by a newline), so the retriever always sees the original query terms alongside the expansion. This design ensures that the pseudo-document augments rather than replaces the original query signal.

**Adaptation to New Corpora.** OPQE can be applied to new corpora or tasks in the same way as DeepRetrieval: by training on a target corpus with retrieval-based rewards from the corresponding retriever–corpus environment. Since OPQE operates on the query side and treats the retriever as a black box, it is agnostic to corpus updates — the retriever index can be updated independently without retraining the policy. When per-corpus RL training is infeasible, the zero-shot SPQE baseline remains applicable.

## 5.2 Experiment, Result and Analysis

**Experimental Setup.** We experiment on evidence-seeking retrieval and ad hoc retrieval datasets. Our implementation is the same as Section 4.2 except for the prompt template to instruct the policy model to write a pseudo-document.

**Main Results.** We report the results of our hybrid approach in Table 4. The key finding is that our proposed method, OPQE, successfully integrates the strengths of both prompting and on-policy optimization, achieving the best overall performance, especially in the dense retrieval setting.

For dense retriever, the benefits of our hybrid approach are most apparent. On the ad hoc retrieval results, the OPQE-7B model achieves the best average score of 58.1 in our setting, compared to the standard DR-7B (DeepRetrieval) model (57.5) and the prompt-based SPQE (56.6). This improvement is consistent across multiple datasets, with OPQE improving results on benchmarks like DL20 (70.5). Similarly, on the evidence-based retrieval datasets, the OPQE-3B model increases the average performance to 80.2, improving over other methods under the same retriever and training setup. This suggests that using RL to fine-tune pseudo-document generation can produce query representations that are effective for dense retrievers.

For the sparse BM25 retriever, the results show that OPQE is highly competitive. While the zero-shot SPQE remains the top performer on evidence-based datasets (78.2), our OPQE-7B model closes the gap significantly and improves over the standard DR-7B model (76.5 vs. 75.6). On the ad hoc retrieval datasets, the OPQE-7B model matches the top performance of SPQE, with both achieving an average score of 51.6. In summary, by framing on-policy optimization as generating a pseudo-document rather than a rewritten query, our OPQE method combines the generative structure of prompting with the targeted fine-tuning of reinforcement learning, and yields strong performance across benchmarks.

We caution, however, against over-interpreting these results. What our experiments establish is that, under a fixed on-policy regime (the same backbone, training setup, and reward), optimizing over document-like outputs is a more effective formulation than direct query rewriting. They do *not* isolate the causal mechanism behind this gain: as the output-length analysis below makes clear, the pseudo-document formulation also changes the length and information content of the retrieval input, and part of the improvement may be attributable to these factors rather than to the document-style format per se. Fully disentangling them is non-trivial, since output length, structure, and content are inherently coupled in this setting — enforcing strict length or content matching (e.g., via truncation or constrained decoding) would alter the pseudo-document distribution and change the nature of the method itself. We therefore present OPQE as an effective formulation under a fixed RL regime rather than as evidence for any single causal factor, and leave a cleaner separation of these effects to future work.

**Output Length Analysis.** To examine whether performance gains can be attributed to output length alone, we report average output lengths (character count of the generated passage or rewritten query, excluding reasoning tokens) in Table 5. RL-optimized methods produce substantially shorter outputs than

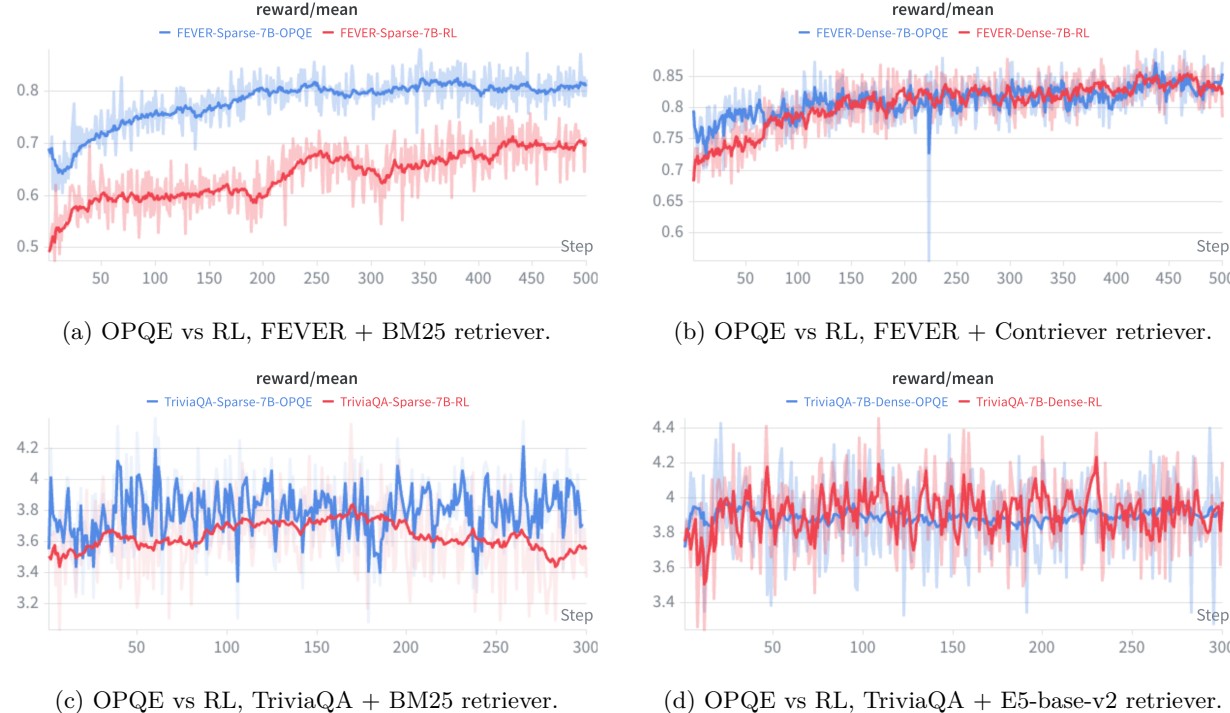

(a) OPQE vs RL, FEVER + BM25 retriever.

(b) OPQE vs RL, FEVER + Contriever retriever.

(c) OPQE vs RL, TriviaQA + BM25 retriever.

(d) OPQE vs RL, TriviaQA + E5-base-v2 retriever.

Figure 1: Reward curves of OPQE vs. RL on FEVER and TriviaQA datasets across different retrievers.

Table 5: Average output length (characters, excluding reasoning tokens) of different augmentation methods on evidence-seeking datasets. DR generates rewritten queries; SPQE and OPQE generate pseudo-documents. Sparse and Dense denote the retriever used as the RL reward signal during training.

|  | NQ | TriviaQA | SQuAD | Avg |
|---|---|---|---|---|
| DR-7B (Sparse) | 61 | 65 | 60 | 62 |
| DR-7B (Dense) | 91 | 129 | 110 | 110 |
| OPQE-7B (Sparse) | 194 | 96 | 101 | 130 |
| OPQE-7B (Dense) | 108 | 136 | 144 | 129 |
| SPQE (Qwen3-32B) | 630 | 590 | 661 | 627 |

training-free SPQE: DR-7B averages 62–110 characters and OPQE-7B averages 113–130, compared to 627 for SPQE (Qwen3-32B), suggesting that RL compresses outputs to retrieval-relevant content. Additionally, the retriever type influences the learned output length differently for DR and OPQE, indicating that RL adapts the generation strategy to the retrieval signal. Note that the length difference between DR and OPQE is *inherent to the method design* (pseudo-documents vs. rewritten queries); we report these statistics for transparency.

**Training Trajectories of OPQE.** To better understand the performance gains of our hybrid OPQE method, we analyze its reward trajectories during training compared to the standard RL approach in Fig. 1. These plots provide evidence for our hypotheses about the effectiveness of incorporating a pseudo-document generation structure.

First, the trajectories validate our hypothesis that this structure effectively leverages the LLM's internal knowledge, particularly for sparse retrievers. In Fig. 1a and Fig. 1c (FEVER and TriviaQA with BM25), the OPQE policy (blue line) starts with a significantly higher average reward than the standard RL policy (red line). This demonstrates that the initial, zero-shot pseudo-documents are already high-quality query

```
Case Study

Query: who cooks for the president of the united states
# DR response (Qwen2.5-7B)
<think>
The original query is "who cooks for the president of the united states". To expand this
    query, we can add more context and possibly more related information. So, the expanded
    query could be "who cooks for the president of the united states in the white house or
    state banquet"
</think>
<answer>
{
    "query": "who cooks for the president of the united states in the white house or state
    banquet"
}
</answer>

# OPQE response (Qwen2.5-7B)
<think>
The President of the United States is a high-ranking government official. The person who
    cooks for the President is typically a member of the White House staff. The current chef
    for the President of the United States is Cristeta Comerford.
</think>
<answer>
{
    "query": "who cooks for the president of the united states\nThe person who cooks for the
    President of the United States is typically a member of the White House staff. The
    current chef for the President of the United States is Cristeta Comerford."
}
</answer>

# SPQE response (Qwen3-32B)
who cooks for the president of the united states. The President of the United States is
    typically served meals prepared by a team of professional chefs employed by the White
    House. These chefs are part of the White House Executive Chef staff, who are responsible
    for planning, preparing, and presenting meals for the president, their family, and
    official guests. The Executive Chef oversees the kitchen operations and ensures that the
    food meets both culinary and dietary standards. The menu often reflects a blend of
    American cuisine and international dishes, tailored to the president's preferences and
    any official state dinners. The White House kitchen also works closely with nutritionists
    to accommodate the president's health needs. The role of the White House chef is both
    prestigious and demanding, requiring expertise in a variety of culinary styles and an
    understanding of the security and protocol requirements of working in the White House.

# Groundtruth:
['Cristeta Comerford', 'White House Executive Chef', 'The White House Executive Chef']
```

Figure 2: DR, OPQE and SPQE examples on the same query, NQ dataset, sparse retrieval.

expansions that provide rich lexical signals for the term-based BM25 retriever. OPQE essentially begins with a "warm start" by using the LLM's knowledge, while the standard RL method must learn to generate effective query terms from a lower baseline.

Second, for dense retrievers (Fig. 1b and Fig. 1d), the initial reward gap is less pronounced, but OPQE's reward trajectory is consistently stable and slightly higher. We attribute this to the pseudo-document format being better aligned with the dense retriever's input distribution: since dense encoders are trained

on passage-level contrastive objectives, a generated passage provides richer semantic specification than a rewritten query, giving the RL agent a more effective starting structure to optimize.

In essence, the training dynamics demonstrate OPQE's synergy: it leverages a knowledge-rich starting point from prompting and then uses on-policy optimization to fine-tune this already effective generation process, leading to higher and more stable rewards throughout training.

**Case Studies.** We show an example case study in Fig. 2; additional examples are in Section F. In this example, DR appends generic context terms ("white house or state banquet") that have limited term overlap with the ground-truth passage. OPQE, by contrast, generates the exact answer entity ("Cristeta Comerford") and related terms ("White House staff"), directly increasing BM25 term match with relevant documents. SPQE produces a much longer passage with correct key terms ("White House Executive Chef") but also substantial filler text, consistent with the length analysis in Table 5.

## 6 Conclusion and Future Works

In this paper, we systematically evaluated and compared two LLM-based query augmentation paradigms: simple prompting-based query augmentation and RL-based query augmentation, across a variety of IR tasks. Our results suggest that prompt-based methods with advanced LLMs can perform on par with, or even surpass, RL-based methods in many scenarios, while RL-based approaches are particularly effective at enabling smaller LLMs to excel at this task. Furthermore, a hybrid approach, where pseudo-document generation prompts from prompt-based methods are used to provide stronger initialization for RL-based training, achieved the best overall performance. These findings suggest that RL-based query augmentation frameworks can be further optimized through improved task reformulation. Importantly, prompt-based methods like SPQE are directly usable in a zero-shot manner across datasets and retrievers, while on-policy RL approaches (including DeepRetrieval-style query rewriting and OPQE) typically require training a separate policy for each new corpus or task.

While our study highlights the strengths and trade-offs of prompting-based and RL-based query augmentation, several avenues remain open for future exploration. First, developing more lightweight yet effective prompting strategies for zero-shot query augmentation could reduce reliance on large LLMs and broaden applicability in resource-constrained settings.

Second, our study deliberately scopes to on-policy optimization (Section 2), which leaves a controlled comparison against off-policy alternatives open. A natural next step is to compare on-policy PPO against off-policy preference tuning for query generation, e.g., DPO-style alignment with ranking-derived preferences (Rafailov et al., 2023), under matched data and compute budgets, clarifying whether the online retrieval signal is worth its cost relative to learning from pre-collected preference data.

Third, OPQE fixes the generation format (a Wikipedia-style pseudo-document) by hand, motivated by the observation that the most effective format depends on the retriever's training distribution. A promising generalization is to let the policy *learn* the output format itself — jointly optimizing what to generate, how long, and in what style for a given retriever — rather than committing to pseudo-documents a priori. This would turn the retriever-aware format choice from a manual design decision into part of the learned policy.

Fourth, exploring query augmentation prompt design and RL training beyond the classic single-round retrieval paradigm, such as in multi-hop or deep research settings (Chen et al., 2025b), remains an important direction. Since query augmentation is the atomic retrieval action of agentic search, a natural next step is to embed an OPQE-style action within multi-turn or parallel-search agents that decompose a query into sub-queries and aggregate evidence across turns (Zhao et al., 2025b; Ko et al., 2025; Bashir et al., 2026); understanding when a better-optimized single action obviates more elaborate orchestration (and vice versa) is an open question that our controlled single-turn findings can help inform.

A further direction concerns the reward itself: our rewards are *verifiable* retrieval metrics (Recall, NDCG, Completeness) computed against relevance labels, which are clean but unavailable for many realistic, open-ended tasks. A natural extension is to use a performant retriever/reranker/LLM-as-a-Judge as a proxy for relevance (Clarke & Dietz, 2024; Xu, 2024; Farzi & Dietz, 2025; Xu et al., 2025a;b; Zhang et al., 2025).

Another exciting direction is to optimize the retrieval component against *downstream* task performance where the objective is hard to verify — for instance, the quality of a generated research report or long-form answer that the retrieved evidence supports. Recent work on RL beyond verifiable domains, using rubric-based rewards (Gunjal et al., 2025) and co-evolving rubrics for deep research (Shao et al., 2025a), suggests a path: rubric- or judge-derived signals over the final output could provide gradient to the retrieval action even when no relevance labels exist, tying query augmentation directly to end-task utility rather than a proxy retrieval metric.

Finally, both DeepRetrieval-style rewriting and OPQE generate an explicit reasoning trace before the query, and a natural question is whether that trace faithfully reflects how the policy arrives at its augmentation. As query-augmentation policies are increasingly optimized with outcome-based RL and embedded as actions in larger agents, the reasoning they emit becomes a tempting target for oversight — but only if it is faithful and monitorable rather than post-hoc rationalization (Turpin et al., 2023; Bentham et al., 2024; Guan et al., 2025). Studying the faithfulness of reasoning traces in retrieval policies, and whether reward signals that explicitly credit faithful reasoning (Xu et al., 2026b) can be applied to query augmentation without sacrificing retrieval effectiveness, is an avenue we find particularly compelling.

## Limitations and Ethical Risks

Our on-policy RL experiments focus on PPO and use lightweight policies ($<$7B), due to compute constraints. In our experiments, RL policies are trained separately per dataset, and we do not study cross-domain transfer following DeepRetrieval's formulation; in practice, re-training a dedicated policy for each new corpus/task can be a bottleneck compared to zero-shot prompting baselines. Like other retrieval-optimized training setups, the RL results can be sensitive to reward design and prompt templates, and may require careful tuning when transferring to new retrievers, corpora, or evaluation metrics. Finally, as discussed in Section 5.2, OPQE's gains over direct rewriting should not be attributed to any single factor such as output length, since its formulation inherently couples output structure, length, and content.

This work is based on public benchmarks, but tool retrieval benchmarks may contain overlapping or near-duplicate tools across sources; such overlap can amplify gains from query expansion and may not fully reflect deployment settings with frequently updated tool inventories. We do not introduce new user-facing systems, and we do not anticipate additional ethical risks.

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

## A   Dataset Statistics and Licenses

We report statistics of each dataset in Table 6. Most datasets are released under permissive open licenses such as CC BY-SA (Natural Questions, SQuAD, HotpotQA, FEVER), CC BY (MS MARCO, DL19, DL20, SciFact), and Apache 2.0 (TriviaQA), while NFCorpus and FiQA restrict usage to academic or non-commercial purposes. SciFact uses ODC-By 1.0 for its corpus. ToolRet is a composite benchmark repurposed from multiple sources with mixed licensing; details are provided in Appendix B and Table 6 of Shi et al. (2025).

Table 6: Detailed dataset statistics.

| Dataset | # Train | # Val | # Test | Corpus Size |
|---|---|---|---|---|
| *Evidence-based Retrieval Datasets* | | | | |
| Natural Questions | 79,168 | 8,757 | 3,610 | 21M |
| TriviaQA | 78,785 | 8,837 | 11,313 | 21M |
| SQuAD | 87,599 | – | 10,570 | 21M |
| *Ad Hoc Retrieval Datasets* | | | | |
| MS-MARCO | 502,939 | – | 5,360 | 8.84M |
| DL19 | – | – | 43 | 8.84M |
| DL20 | – | – | 54 | 8.84M |
| NFCorpus | 2,590 | 647 | 323 | 3.6K |
| HotpotQA | 85,000 | 12,852 | 7,405 | 5.23M |
| FEVER | 109,810 | 13,332 | 6,666 | 5.42M |
| SciFact | 818 | 440 | 339 | 5K |
| FIQA | 5,500 | 500 | 648 | 57K |
| *Tool Retrieval Dataset* | | | | |
| ToolRet-Train | 208,826 | – | – | 31,123 |
| ToolRet-Web | – | – | 4,916 | 36,978 |
| ToolRet-Code | – | – | 950 | 3,794 |
| ToolRet-Customized | – | – | 1,749 | 2,443 |

## B    Results of SPQE with Different LLMs

Table 7: Performance (Hit@20) of SPQE with different LLMs on evidence-seeking datasets.

| | Base | GPT-4o-mini | Qwen3-32B | GPT-OSS-120B |
|---|---|---|---|---|
| *Sparse BM25 Retriever* | | | | |
| NQ | 63.0 | 80.7 | 77.6 | 81.1 |
| TriviaQA | 76.4 | 84.1 | 81.6 | 84.5 |
| Squad | 71.1 | 69.7 | 70.3 | 74.3 |
| Average | 70.2 | 78.2 | 76.5 | 80.0 |
| *Dense E5-base-v2 Retriever* | | | | |
| NQ | 86.1 | 85.2 | 83.9 | 85.1 |
| TriviaQA | 83.5 | 85.0 | 83.2 | 85.5 |
| Squad | 70.2 | 68.8 | 68.5 | 68.5 |
| Average | 79.9 | 79.7 | 78.5 | 79.7 |

We report performance comparison of SPQE with different LLMs in Table 7, Table 8, Table 9 and Table 10, Table 11, respectively.

## C    Reward Formulations

The reward function $r(q')$ we used in DeepRetrieval's RL training consists of format reward $r_{\text{format}}$ and retrieval reward $r_{\text{retrieval}}$, where $r_{\text{format}}$ evaluates whether the trained policy follows the required response format and $r_{\text{retrieval}}$ evaluates the retrieval performance of the augmented query. The formulation is $r(q') = r_{\text{format}}(q') \cdot r_{\text{retrieval}}(q')$ where the format reward is an indicator function:

$$r_{\text{format}}(q') = \begin{cases} 0, & q' \text{ violates the required format} \\ 1, & q' \text{ follows the required format} \end{cases}$$

For retrieval reward, we report the specific instantiation for each task in Table 12.

Table 8: Performance of backbone-matched SPQE on ad hoc retrieval datasets. Sparse uses BM25 retriever; dense uses Contriever-msmarco retriever.

| | Base | | SPQE (Qwen2.5-3B-Instruct) | | SPQE (Qwen2.5-7B-Instruct) | |
|---|---|---|---|---|---|---|
| | NDCG@10 | Recall@100 | NDCG@10 | Recall@100 | NDCG@10 | Recall@100 |
| *Sparse BM25 Retriever* | | | | | | |
| MSMARCO | 22.8 | 65.8 | 19.7 | 63.3 | 19.1 | 63.3 |
| DL19 | 50.6 | 49.7 | 53.8 | 54.1 | 57.3 | 57.4 |
| DL20 | 48.0 | 56.7 | 51.7 | 61.0 | 53.4 | 61.7 |
| NFCorpus | 32.2 | 24.6 | 31.6 | 30.9 | 31.6 | 30.1 |
| HotpotQA | 63.3 | 79.6 | 56.4 | 77.3 | 57.4 | 78.7 |
| FEVER | 65.1 | 91.8 | 73.2 | 93.1 | 74.2 | 93.0 |
| SciFact | 67.9 | 92.5 | 69.3 | 94.4 | 69.7 | 96.3 |
| FiQA | 23.6 | 53.9 | 17.3 | 47.3 | 19.0 | 50.0 |
| Avg | 46.7 | 64.3 | 46.6 | 65.2 | 47.7 | 66.3 |
| *Dense Contriever-msmarco Retriever* | | | | | | |
| MSMARCO | 40.7 | 89.1 | 30.3 | 79.7 | 30.0 | 80.1 |
| DL19 | 67.5 | 61.3 | 61.3 | 62.3 | 64.1 | 63.4 |
| DL20 | 66.6 | 71.5 | 62.3 | 66.8 | 65.1 | 72.0 |
| NFCorpus | 32.8 | 30.1 | 32.7 | 34.4 | 34.3 | 33.1 |
| HotpotQA | 63.8 | 77.7 | 59.5 | 76.9 | 60.1 | 77.7 |
| FEVER | 75.8 | 94.9 | 81.2 | 95.0 | 82.3 | 95.5 |
| SciFact | 67.7 | 94.7 | 66.3 | 97.0 | 67.5 | 97.0 |
| FiQA | 32.9 | 65.6 | 24.0 | 58.7 | 28.1 | 63.9 |
| Avg | 56.0 | 73.1 | 52.2 | 71.4 | 53.9 | 72.8 |

## D  Completeness Metric

Let $G(q)$ be the set of ground-truth documents for query $q$, and $R_k(q)$ be the set of top-$k$ retrieved documents for query $q$. Then completeness@$k$ is defined as:

$$\text{Completeness@}k(q) = \begin{cases} 1, & \text{if } G(q) \subseteq R_k(q), \\ 0, & \text{otherwise.} \end{cases}$$

Averaged over a query set $Q$, we have

$$\text{Completeness@}k = \frac{1}{|Q|} \sum_{q \in Q} \mathbf{1}[\, G(q) \subseteq R_k(q) \,].$$

## E  Prompt Templates

We show RL prompt templates in Fig. 3 and Fig. 4, SPQE prompt template in Fig. 5 and OPQE prompt template in Fig. 6.

## F  Case Studies

We present additional case studies comparing DR, OPQE, and SPQE outputs on NQ queries with BM25 retrieval.

In Fig. 7, DR adds "this year" to the query but does not introduce any location-specific terms, providing little additional signal for BM25. OPQE generates the correct answer ("Beijing, China") as a concise pseudo-document, directly adding high-value terms. SPQE produces detailed text about the 2026 Milan-Cortina

Table 9: Performance of SPQE with stronger LLMs (computed-matched versus DeepRetrieval) on ad hoc retrieval datasets. Sparse uses BM25 retriever; dense uses Contriever-msmarco retriever.

| | SPQE (GPT-4o-mini) | | SPQE (Qwen3-32B) | | SPQE (GPT-OSS-120B) | |
|---|---|---|---|---|---|---|
| | NDCG@10 | Recall@100 | NDCG@10 | Recall@100 | NDCG@10 | Recall@100 |
| *Sparse BM25 Retriever* | | | | | | |
| MSMARCO | 21.9 | 67.5 | 20.3 | 66.6 | 21.9 | 67.3 |
| DL19 | 60.0 | 58.8 | 60.9 | 59.9 | 58.0 | 57.6 |
| DL20 | 60.1 | 68.5 | 59.3 | 66.9 | 57.0 | 65.0 |
| NFCorpus | 33.9 | 30.8 | 34.4 | 31.0 | 33.0 | 32.1 |
| HotpotQA | 66.5 | 84.8 | 61.3 | 81.8 | 69.3 | 85.9 |
| FEVER | 79.8 | 95.3 | 77.9 | 94.8 | 78.3 | 95.7 |
| SciFact | 71.4 | 95.9 | 71.2 | 95.9 | 73.8 | 96.4 |
| FiQA | 19.5 | 50.0 | 21.3 | 54.6 | 19.2 | 49.4 |
| Avg | 51.6 | 68.9 | 50.8 | 68.9 | 51.3 | 68.7 |
| *Dense Contriever-msmarco Retriever* | | | | | | |
| MSMARCO | 31.9 | 81.5 | 31.7 | 81.6 | 28.2 | 78.2 |
| DL19 | 70.1 | 65.6 | 69.6 | 65.7 | 64.6 | 61.2 |
| DL20 | 69.3 | 74.0 | 66.8 | 72.8 | 57.3 | 68.3 |
| NFCorpus | 34.3 | 33.5 | 35.0 | 32.7 | 29.1 | 28.9 |
| HotpotQA | 68.2 | 83.7 | 62.8 | 79.9 | 66.9 | 82.7 |
| FEVER | 83.9 | 96.0 | 83.2 | 95.8 | 80.6 | 95.0 |
| SciFact | 68.9 | 95.7 | 69.5 | 96.3 | 67.3 | 92.9 |
| FiQA | 26.5 | 60.9 | 30.7 | 66.5 | 20.4 | 53.5 |
| Avg | 56.6 | 73.9 | 56.2 | 73.9 | 51.8 | 70.1 |

Table 10: Performance of backbone-matched SPQE on ToolRetrieval datasets. Sparse uses BM25 retriever; dense uses E5-base-v2 retriever.

| | Base | | SPQE (Qwen2.5-3B-Instruct) | | SPQE (Qwen2.5-7B-Instruct) | |
|---|---|---|---|---|---|---|
| | Recall@10 | Completeness@10 | Recall@10 | Completeness@10 | Recall@10 | Completeness@10 |
| *Sparse BM25 Retriever* | | | | | | |
| ToolRet-Web | 33.6 | 26.0 | 39.2 | 30.8 | 38.2 | 30.1 |
| ToolRet-Code | 44.3 | 37.1 | 44.0 | 41.0 | 46.5 | 43.7 |
| ToolRet-Customized | 41.7 | 24.9 | 45.7 | 29.1 | 47.1 | 30.1 |
| Average | 39.9 | 29.4 | 43.0 | 33.6 | 43.9 | 34.6 |
| *Dense E5-base-v2 Retriever* | | | | | | |
| ToolRet-Web | 41.1 | 31.6 | 45.9 | 36.0 | 45.2 | 35.3 |
| ToolRet-Code | 33.4 | 30.5 | 40.2 | 37.0 | 41.1 | 38.0 |
| ToolRet-Customized | 41.8 | 26.8 | 45.1 | 29.9 | 45.6 | 29.5 |
| Average | 38.8 | 29.6 | 43.7 | 34.3 | 43.9 | 34.3 |

games, which contains "Beijing" only as a passing mention — illustrating how verbose generation can dilute key terms.

In Fig. 8, DR rephrases the query ("what authority sets up") without adding factual content. OPQE generates the answer ("Congress") along with explanatory context about administrative independence. SPQE again produces a comprehensive but lengthy passage that includes the answer term, together with tangential details about specific agencies.

Table 11: Performance of SPQE with stronger LLMs on ToolRetrieval datasets. Sparse uses BM25 retriever; dense uses E5-base-v2 retriever.

| | SPQE (GPT-4o-mini) | | SPQE (Qwen3-32B) | | SPQE (GPT-OSS-120B) | |
|---|---|---|---|---|---|---|
| | Recall@10 | Completeness@10 | Recall@10 | Completeness@10 | Recall@10 | Completeness@10 |
| *Sparse BM25 Retriever* | | | | | | |
| ToolRet-Web | 38.0 | 30.2 | 38.1 | 29.8 | 39.0 | 30.5 |
| ToolRet-Code | 46.0 | 42.7 | 47.4 | 44.4 | 46.8 | 43.0 |
| ToolRet-Customized | 49.4 | 32.4 | 46.7 | 30.1 | 47.1 | 31.4 |
| Average | 44.4 | 35.1 | 44.1 | 34.8 | 44.3 | 34.9 |
| *Dense E5-base-v2 Retriever* | | | | | | |
| ToolRet-Web | 45.1 | 35.0 | 45.7 | 35.6 | 44.6 | 34.6 |
| ToolRet-Code | 42.2 | 38.3 | 44.9 | 41.9 | 43.1 | 40.4 |
| ToolRet-Customized | 47.2 | 31.2 | 45.6 | 29.6 | 45.2 | 29.4 |
| Average | 44.8 | 34.8 | 45.4 | 35.7 | 44.3 | 34.8 |

Table 12: Retrieval reward function $r_{\text{retrieval}}(q')$.

| Task | Retrieval Reward |
|---|---|
| Evidence-seeking Retrieval | 5.0 if rank $\leq 5$
4.0 if rank $\leq 20$
2.0 if rank $\leq 50$
1.0 if rank $\leq 100$
0.5 if rank $\leq 1000$
0.1 if rank $\leq 3000$
-2.5 otherwise |
| Ad hoc Retrieval | NDCG@10 score |
| Tool Retrieval | Completeness@10 score |

**RL (DeepRetrieval) Prompt Template**

```
<|im_start|>system
You are a helpful assistant. You first thinks about the reasoning process in the mind and
    then provides the user with the answer.
<|im_end|>

<|im_start|>user
You are a query rewriting expert. Your task is to create query terms for user query to find
    relevant literature in a Wikipedia corpus using BM25.

Show your reasoning inside the <think> </think> tags.
Your final output must be inside the <answer> </answer> tags, and strictly in JSON format.
For example:

<think>
[reasoning process]
</think>
<answer>
{
    "query": "...."
}
</answer>
Note: The query should use Boolean operators (AND, OR) and parentheses for grouping terms
    appropriately.

Here is the user query:
[USER_QUERY]

Assistant: Let me rewrite the query with reasoning.
<think>
```

Figure 3: The RL (DeepRetrieval) prompt template for NQ dataset—sparse retrieval. Notice the boolean operators.

---

**RL (DeepRetrieval) Prompt Template**

```
<|im_start|>system
You are a helpful assistant. You first thinks about the reasoning process in the mind and
    then provides the user with the answer.
<|im_end|>

<|im_start|>user
You are an expert in generating queries for dense retrieval. Given a web search query, your
    task is to retain the original query while expanding it with additional semantically
    relevant information, to retrieve relevant passages that answer the query. If no useful
    expansion is needed, return the original query as is.

Show your reasoning inside the <think> </think> tags.
Your final output must be inside the <answer> </answer> tags, and strictly in JSON format.
For example:

<think>
[reasoning process]
</think>
<answer>
{
    "query": "...."
}
</answer>

Here is the user query:
[USER_QUERY]

Assistant: Let me rewrite the query with reasoning.
<think>
```

Figure 4: The RL (DeepRetrieval) prompt template for NQ dataset — dense retrieval.

---

**SPQE Prompt Template**

```
Write a short Wikipedia passage to answer this question.
Question: [USER_QUERY]
Passage:
```

Figure 5: SPQE prompt template for NQ dataset. We use same prompt template for sparse and dense retrieval.

---

**OPQE Prompt Template**

```
<|im_start|>system
You are a helpful assistant. First think through the reasoning internally, then provide the
    answer as instructed.
<|im_end|>

<|im_start|>user
You are a helpful assistant. Given a search query, write a concise Wikipedia-style passage (a
    few sentences) that answers the query. This passage will then be concatenated with the
    original query to form an enhanced query.

Show your reasoning inside the <think> </think> tags.
Your final output must be inside the <answer> </answer> tags, and strictly in JSON format.
For example:

<think>
[reasoning process]
</think>
<answer>
{
    "query": "original query. concise passage"
}
</answer>

Note: Make sure you write the passage instead of directly answering the question.

Here is the user query:
[USER_QUERY]

Assistant: Let me augment the query with reasoning.
<think>
```

Figure 6: The OPQE prompt template for NQ dataset. We use same prompt template for sparse and dense retrieval.

---

**Additional Case Study I**

```
Query: where are the winter olympic games being played
# DR response (Qwen2.5-7B)
<think>
The original query "where are the winter olympic games being played" seems clear but could
    benefit from additional information to make it more specific or searchable. Winter
    Olympic games are typically held in specific cities or venues, so specifying "this year"
    might help narrow down the location.
</think>
<answer>
{
    "query": "where are the winter olympic games being played this year"
}
</answer>

# OPQE response (Qwen2.5-7B)
<think>
The Winter Olympic Games 2022 are being played in Beijing, China.
</think>
<answer>
{
    "query": "where are the winter olympic games being played\nThe Winter Olympic Games 2022
    are being played in Beijing, China."
}
</answer>

# SPQE response (Qwen3-32B)
where are the winter olympic games being played. The 2026 Winter Olympic Games are scheduled
    to be held in Milan-Cortina d'Ampezzo, Italy, from 6 to 22 February 2026. This will mark
    the first time since Cortina dAmpezzo hosted the Games in 1956 that the Winter Olympics
    return to Italy. Competitions will take place across multiple venues in the Lombardy and
    Veneto regions, including alpine events in the Dolomites. The 2026 edition follows the
    2022 Games in Beijing, China, and precedes the 2030 Games, tentatively awarded to Salt
    Lake City, Utah, and Revelstoke, British Columbia (joint bid). The Milan-Cortina hosts
    aim to showcase sustainable infrastructure and highlight Northern Italys alpine heritage.

# Groundtruth:
['Beijing']
```

Figure 7: DR, OPQE and SPQE examples on the same query, NQ dataset, sparse retrieval.

---

**Additional Case Study II**

```
Query: who sets up the independent agencies of the executive branch
# DR response (Qwen2.5-7B)
<think>
The user is asking about the executive branch. Specifically, they want to know which
    authority sets up the independent agentics of the executive branch. Therefore, I can
    revise the query to "what authority sets up the independent agencies of the executive
    branch" to make it more specific.
</think>
<answer>
{
    "query": "what authority sets up the independent agencies of the executive branch"
}
</answer>

# OPQE response (Qwen2.5-7B)
<think>
Independent agencies of the United States executive branch are created by Congress and given
    independence from day-to-day administrative control. The President of the United States
    appoints the heads of these agencies, but they have a degree of independence from the
    executive branch.
</think>
<answer>
{
    "query": "who sets up the independent agencies of the executive branch\nIndependent
    agencies of the United States executive branch are created by Congress and given
    independence from day-to-day administrative control. The President of the United States
    appoints the heads of these agencies, but they have a degree of independence from the
    executive branch."
}
</answer>

# SPQE response (Qwen3-32B)
The independent agencies of the United States executive branch are established by acts of
    Congress through legislation, granting them authority to oversee specialized regulatory,
    research, or public service functions. While they operate independently of the White
    House in terms of management, the President appoints agency heads (subject to Senate
    confirmation) to lead these entities. Examples include the Environmental Protection
    Agency (EPA), Centers for Disease Control and Prevention (CDC), and National Institute of
    Standards and Technology (NIST). Some independent agencies, like the Office of National
    Drug Control Policy, are structured as part of federal departments (e.g., the Department
    of Health and Human Services), while others function autonomously. These agencies are
    designed to balance administrative flexibility with legislative mandates, ensuring
    technical expertise is applied to complex policy areas without daily White House
    oversight.

# Groundtruth:
['Congress']
```

Figure 8: DR, OPQE and SPQE examples on the same query, NQ dataset, sparse retrieval.

