# OpenReview forum: "Rethinking On-policy Optimization for Query Augmentation"
_TMLR — Accepted by TMLR_

### Review · Reviewer_Bxbg · 2026-03-09

**Summary Of Contributions:**

This paper studies query augmentation for information retrieval using large language models. The authors conduct a systematic comparison between two main paradigms: prompting-based pseudo-document generation and reinforcement learning (RL)–based query rewriting. Under a compute-aware comparison, the authors find that simple prompting-based query expansion can often match or outperform RL-trained policies, particularly when stronger LLMs are used for inference.

Motivated by this finding, the paper proposes On-policy Pseudo-document Query Expansion (OPQE), a hybrid approach that applies on-policy RL to pseudo-document generation rather than direct query rewriting. OPQE integrates the generative structure of prompting with retrieval-aware optimization via RL rewards.

Experiments across multiple retrieval tasks—including evidence-seeking retrieval, ad hoc retrieval, and tool retrieval—show that OPQE consistently achieves strong performance and often surpasses both standalone prompting and RL-based rewriting baselines.

**Strengths**

1. The paper provides a controlled comparison between prompting-based query expansion and RL-based query rewriting, which is currently missing in the literature.
2. The finding that simple prompting can rival RL-based approaches under realistic compute assumptions is useful for practitioners.
3. The proposed OPQE method is conceptually simple yet effective, reframing RL optimization toward pseudo-document generation.

**Weaknesses**

1. The paper does not deeply analyze why pseudo-document generation helps (e.g., term distribution, length effects, embedding alignment). The authors could add case studies, e.g., the analysis of generated pseudo-documents and their influence on the retrieval performance. why they have better performance on BM25 or Dense retrieval, and what contents or key words/sentences are critical for the improvement of the retrieval performance in a single case.

2. There exists prior work that reports a similar observation regarding the interaction between query rewriting and dense retrievers (see: https://arxiv.org/pdf/2503.24289, Appendix E.1.3). That paper shows that directly rewriting a query using a standard prompting template can perform poorly when used with dense retrievers. By analyzing the pretraining setup of the dense retriever (which was trained using user review–product pairs via contrastive learning), the authors found that rewriting the query into a review-style text significantly improves retrieval performance.
The authors should discuss this related work and clarify how their findings differ from or relate to this prior observation.

**Audience:**

Yes

**Audience Explanation:**

The paper addresses an important question at the intersection of LLMs, reinforcement learning, and information retrieval. The empirical finding that training-free prompting can rival RL-based approaches under realistic compute settings is practically relevant and may influence how future systems are designed. Additionally, the proposed hybrid OPQE method offers a simple modification to existing RL pipelines that yields improved results.
Given the growing interest in LLM-based retrieval, RAG pipelines, and tool-use systems, the insights and experimental findings presented in this paper would likely be of interest to researchers and practitioners in TMLR’s audience.

**Claims And Evidence:**

Yes

**Claims Explanation:**

The experimental setup largely follows the DeepRetrieval framework and includes multiple datasets and retrievers. The paper also evaluates both sparse and dense retrieval models and extends the study to tool retrieval tasks, which strengthens the empirical evidence.

**Requested Changes:**

I suggest the authors consider the following changes.

1. Add case studies or qualitative analysis
The paper would benefit from including several case studies of generated queries or pseudo-documents, as stated in the weakness above. Such qualitative analysis would strengthen the paper’s insights beyond purely quantitative results.
2. Add discussion of related work
The authors should discuss the following related work in the Related Work section: https://arxiv.org/pdf/2503.24289, as mentioned in weakness above.
3. Minor issues and suggestions
- Page 5, Section 4.1: zero-shot method HyDE Gao et al. (2023) -> zero-shot method (HyDE Gao et al. (2023)).
- Page 8: train corpus → training corpus, trainset → training set
- Page 6 mentions that RL training requires 7–14× A100 40GB GPUs. This estimate may need to be reconsidered. In practice, training a 3B model often requires roughly 4× A100 40GB, while a 7B model may require around 8× A100 40GB, depending on the training setup. The authors should double-check and clarify these numbers.
- Page 10, Figure 1: The reward curves are somewhat noisy. Applying smoothing (e.g., moving average) could make the training trends clearer and easier to interpret.

---

### Review · Reviewer_LCfh · 2026-04-02

**Summary Of Contributions:**

This paper studies LLM-based query augmentation for black-box retrieval settings and makes two main contributions. First, it conducts a compute-aware comparison of prompting-based pseudo-document expansion (SPQE) against on-policy RL-based query rewriting (DeepRetrieval-style PPO) across evidence-seeking, ad hoc, and tool retrieval tasks, and finds that SPQE with a powerful LLM often matches or surpasses RL-trained smaller policies, particularly for sparse retrieval. Second, it proposes OPQE, which applies on-policy RL to pseudo-document generation rather than direct query rewriting, combining the generative structure of prompting with retrieval-based reward optimization. Experiments show that OPQE outperforms both SPQE and DeepRetrieval, achieving the strongest overall results, with the most pronounced gains in dense retrieval settings.

**Audience:**

Yes

**Audience Explanation:**

Even though I am not yet convinced by all of the claims, I do think the paper is interesting. The comparison between query-like generation and document-like generation is important, and the attempt to evaluate that distinction both with and without RL is valuable. In particular, the paper explores an under-studied direction—applying on-policy RL to pseudo-document generation—and that is a useful contribution even if the evidence needs to be tightened.

I also think the paper has practical relevance. Many realistic retrieval settings do treat the retriever as a black box, and in such settings query-side intervention is the natural design space. The broad empirical coverage across evidence-seeking retrieval, ad hoc retrieval, and tool retrieval also increases the potential interest of the work. So while I do not think the current version fully supports its strongest claims, I do think there is something here that many readers in the area would want to learn from.

**Claims And Evidence:**

No

**Claims Explanation:**

I think the paper contains an interesting empirical story, but there are important gaps between the claims and the evidence.

1. Section 4: the comparison between training-free pseudo-document generation and RL-based query rewriting is not sufficiently controlled

A central claim in Section 4 is that simple, training-free pseudo-document prompting can match or even outperform RL-based query rewriting. I do not find this comparison fully convincing as currently presented. The paper explicitly says that this is a “compute-aware” comparison and also explicitly notes that the SPQE results are not backbone-matched to DR-3B/7B, since stronger inference-time models are used for SPQE, while RL is trained with smaller base policies. The paper also notes that preliminary same-backbone SPQE runs were “not as competitive,” but these results are not reported. This makes it difficult to know how much of the reported advantage comes from the pseudo-document strategy itself versus from model-scale differences.

More importantly, I am not convinced that the paper’s notion of “compute-aware” comparison is itself sufficient to justify the claim. Training cost and inference cost are different resources and matter differently in deployment. RL has an upfront training cost that may be amortized over many future queries, whereas prompting with a stronger LLM incurs per-query inference cost indefinitely. Because these axes are not separated, I do not think the current evidence supports a broad conclusion such as “training-free pseudo-document prompting is stronger than RL-based query augmentation” without substantially more qualification. At minimum, the claims in Section 4 should be reduced, or the missing same-backbone SPQE baselines should be reported.

2. Section 4 / Discussion: the dense-retrieval explanation is not convincing as written

I also found the paper’s explanation for why pseudo-document generation helps dense retrieval to be conceptually weak. The paper suggests that pseudo-document expansion may help because it makes retrieval more “symmetric,” closer to a document-document matching problem. I do not find this explanation persuasive. Historically, one of the main motivations for dense retrieval is precisely that it can model asymmetric query-document relationships beyond surface lexical overlap. Because of that, explaining the gains in terms of “making the task more symmetric” feels under-justified and potentially misleading.

To me, the current evidence is more consistent with alternative explanations, such as:

- richer semantic specification of an otherwise under-specified query,
- better alignment with the dense retriever’s input distribution,
- or simply longer / more information-rich retrieval inputs.

These alternatives are not adequately disentangled in the current draft. I would therefore encourage the authors either to substantially tone down the “symmetry” explanation or to support it with much stronger evidence. As written, this part reads more like a speculative post-hoc story than a well-supported argument.

3. Section 5: the RL-vs-RL comparison is much better, but output-length confounding remains

Here the authors do perform a much more relevant comparison for their core scientific question: under an on-policy RL setting, is it better to optimize direct query rewriting or pseudo-document generation? This comparison is substantially better controlled: it uses the same lightweight Qwen2.5-3B/7B policies, the same overall RL setting, and the same retrieval/training setup except for the prompt formulation.

However, even this comparison is still not fully controlled, because OPQE changes the output structure and likely the length distribution of the generated retrieval input. The paper itself acknowledges that OPQE changes the output structure and typical length distribution relative to direct rewriting. This matters a great deal. If pseudo-document outputs are substantially longer than rewritten queries, then the observed gains may be partly due to producing longer and more information-rich strings, rather than to the pseudo-document format per se. In that case, the claimed conclusion would need to be weakened from “pseudo-document generation is a better RL formulation than query rewriting” to something closer to “under this prompt/output regime, RL over longer document-like retrieval inputs works better.”

I therefore think the paper needs to explicitly report output-length statistics and ideally include a length-controlled or budget-controlled analysis. For example, the authors could report mean/median/percentile token counts for both the generated outputs and the final strings passed to the retriever, and then run a comparison where these distributions are made more comparable (e.g., prompt adjustments, truncation, or a matched subset). Since the paper already appeals to “compute-aware” reasoning, it is especially important that the test-time compute budget also be treated carefully here.

**Requested Changes:**

## Report backbone-matched SPQE baselines using the same Qwen2.5-{3B,7B}-Instruct models.

The current Section 4 comparison mixes method differences with model-scale differences. The paper explicitly frames this as a “compute-aware” comparison and uses stronger inference-time LLMs for SPQE than for the RL-trained DR policies. It also states that preliminary same-backbone SPQE runs were “not as competitive,” but those results are not shown. These missing baselines are essential for a fair comparison. Without them, it is difficult to tell whether the reported gains are due to pseudo-document prompting itself or to using stronger test-time models. Please report these same-backbone SPQE results, at least in the appendix, and discuss them explicitly.

## Add output-length statistics for DR vs OPQE, and perform a length-controlled comparison.

Section 5 is the paper’s most relevant controlled comparison because it keeps the RL setup, policy family, datasets, and retrievers aligned, changing primarily the generation strategy. However, the paper itself acknowledges that OPQE changes the output structure and typical length distribution relative to direct query rewriting. This is a major confound. If OPQE produces substantially longer retrieval inputs, then part of the gain may come from allocating more test-time generation budget, rather than from the pseudo-document strategy itself. Please report:
- mean / median / percentile token lengths of generated outputs,
- mean / median / percentile lengths of the final strings passed to the retriever, and ideally a length-controlled or budget-controlled ablation (e.g., truncation, prompt adjustment, or matched subsets).
This is important both scientifically and from the paper’s own compute-aware perspective.

## Rework the dense-retrieval explanation.

The explanation that pseudo-document expansion helps dense retrieval by making the task more “symmetric” is not sufficiently justified and is conceptually questionable. The authors should either:
(a) provide direct evidence for this claim, or
(b) tone it down and discuss alternative explanations such as richer semantic specification, reduced query underspecification, or better alignment with the retriever’s input distribution.

---

### Review · Reviewer_ETQU · 2026-04-07

**Summary Of Contributions:**

This paper focuses on query augmentation for information retrieval. In particular, it first performs a controlled comparison of DeepRetrieval against a simple, zero-shot prompting baseline (pseudo document generation). It reveals that the simple, training-free prompting method often matches or surpasses more complex RL-trained methods (RL-based query rewriting). Based on this observation, the paper proposes OPQE (On-policy Pseudo-document Query Expansion) , a hybrid method that combines the generative structure of prompting with the targeted optimization of on-policy RL by training the policy to generate pseudo-documents under retrieval-based rewards, and it yields the strongest overall retrieval performance across our benchmarks.

Pros:
1. The paper is well-structured and easy to follow the main claims.
2. The comparison between prompting-based methods and on-policy RL methods seems comprehensive with loads of empirical justifications.
3. The proposed method OPQE demonstrates good performance.

Cons:
1. While it is meaningful to perform a thorough comparison among different query augmentation methods, the benefits and shortcomings of prompting-based methods and RL-based methods are well-known in the IR community. Besides, it is also well-studied that spare retrieval and dense retrieval may perform drastically differently given the same query expansion.
2. The comparison is not fair enough given that different models are used for prompting-based methods and RL-based methods. It is advised that smaller models would be more cost-efficient in the long-run, even though it may take some time to train. In consideration of these differences, the results presented seem not very convincing.
3. The proposed method lacks many details. It is advised to pay more attention to the proposed method, what would be the challenges to train a policy model for pseudo document generation? How can we adapt this method to new documents that may be added to the collection? From my perspective, the new method is more important than the so-called controlled study of existing methods.

**Audience:**

Yes

**Audience Explanation:**

Query augmentation is an important topic in information retrieval and retrieval-augmented generation.

**Broader Impact Concerns:**

The paper has included an ethical risks section.

**Claims And Evidence:**

Yes

**Claims Explanation:**

The results support the main claims in general. However, it is noted that LLMs can also generate query rewrites directly rather than pseudo-documents, the paper does not include results of LLM-based query rewriting.

**Requested Changes:**

More details about the proposed method and more empirical studies to prove its effectiveness.

---

### Decision · Action_Editor_y9LW · 2026-05-13

**Recommendation:** Accept with minor revision

**Additional Comments:**

* The term "on-policy" features prominently in the manuscript, and is even present in the title. However, it is never explained why the RL training in this setting needs to be on-policy, as opposed to off-policy. Could one do off-policy RL for the same task? If not, why not? If yes, why do you prefer on-policy? Please add some explanations about this to Sections 2 and 3. Is this distinction so important that "on-policy" has to be in the title, as opposed to, for example, "Rethinking Reinforcement Learning for Query Augmentation"?

* The sentence in the abstract starting with "Motivated by this discovery" has broken syntax: "we introduce a novel hybrid method [...], which [...] the LLM policy learns to generate a pseudo-document [...]"

* Please add more caveats to Section 5, as discussed with Reviewer LCfh.

**Audience:**

Yes

**Audience Explanation:**

All reviewers thought that the paper is interesting for researchers in information retrieval.

**Claims And Evidence:**

Yes

**Claims Explanation:**

The paper provides a controlled comparison between two approaches to query expansion: prompting-based and RL-based. They argue that RL does not help much (but is much more costly to train). The authors then suggest a modification of the RL approach where a model is trained to append a "Wikipedia-style" paragraph to the query (instead of rewriting the query). They argue that this approach outperforms the rest.